# CONTRASIM – A SIMILARITY MEASURE BASED ON CONTRASTIVE LEARNING

## ABSTRACT

Recent work has compared neural network representations via similarity-based analyses, shedding light on how different aspects (architecture, training data, etc.) affect models' internal representations. The quality of a similarity measure is typically evaluated by its success in assigning a high score to representations that are expected to be matched. However, existing similarity measures perform mediocrely on standard benchmarks. In this work, we develop a new similarity measure, dubbed ContraSim, based on contrastive learning. In contrast to common closed-form similarity measures, ContraSim learns a parameterized measure by using both similar and dissimilar examples. We perform an extensive experimental evaluation of our method, with both language and vision models, on the standard layer prediction benchmark and two new benchmarks that we develop: the multilingual benchmark and the image–caption benchmark. In all cases, ContraSim achieves much higher accuracy than previous similarity measures, even when presented with challenging examples.

## 1 INTRODUCTION

Representation learning is a key property in deep neural networks. But how can we assess the similarity of representations learned by two models? A recent line of work is concerned with developing similarity measures and using them to analyze the models' internal representations. Similarity-based analyses may shed light on how different datasets, architectures, etc., change the model's learned representations. For example, a similarity analysis showed that lower layers in different models are more similar to each other, while fine-tuning affects mostly the top layers (Wu et al., 2020).

Various similarity measures have been proposed for comparing representations, among them the most popular ones are based on centered kernel alignment (CKA) (Kornblith et al., 2019) and canonical correlation analysis (CCA) (Hotelling, 1936; Morcos et al., 2018). They all share a similar methodology: given a pair of feature representations *of the same input*, they estimate the similarity between them, without considering other examples. However, they all perform mediocrely on standard benchmarks. Motivated by that, we propose a new learnable similarity measure.

In this paper, we introduce ContraSim, a new similarity measure, based on contrastive learning (CL) (Chen et al., 2020; He et al., 2020). In contrast to prior work, which defines closed-form general-purpose similarity measures, ContraSim is a task-specific learnable similarity measure that uses examples a high similarity (the *positive* set) and examples that have a low similarity (the *negative* set), to train an encoder that maps representations to the space where similarity is measured. In the projected space, we maximize the representation similarity with examples from the positive set, and minimize it with examples from the negative set.

We experimentally evaluate ContraSim on one standard similarity metrics benchmark and two new benchmarks we introduce in this paper, and demonstrate its superiority compared to common similarity measures. First, we use the known layer prediction benchmark (Kornblith et al., 2019), which assesses whether high similarity is assigned to two architecturally-corresponding layers in two models differing only in their weight initialization. Second, in our proposed multilingual benchmark, we assume a multilingual model and a parallel dataset of translations in two languages. A good similarity measure should assign a higher similarity to the (multi-lingual) representations of a sentence in language A and its translation in language B, compared to the similarity of the same sentence in language A and a random sentence in language B. Third, we design the image–caption benchmark, based on a similar idea. Given an image and its text caption, and correspondingly a vision model

and a language model, a good similarity measure should assign a high similarity to representations of the image and its caption, compared to the similarity of the same image and a random caption.

In both of our new benchmarks, we investigate a more challenging scenario, where instead of choosing a random sentence, we retrieve highly similar sentences as confusing examples, using the Facebook AI Similarity Search (FAISS) library (Johnson et al., 2019). While other similarity measures are highly affected by this change, our method maintains a high accuracy with a very small degradation. We attribute this to the highly separable representations that our method learns.

Finally, in all benchmarks, we show that if we change the training procedure of the encoder to only maximize the similarity of similar examples, the projected representations have poor separation, indicating that the CL procedure is a crucial part of the method's success.

In summary, this work makes the following contributions:

- We introduce a new similarity measure – ContraSim. Inspired by contrastive learning, it uses positive and negative sets to train an encoder that maps representations to the space where similarity is measured.

- We propose two new benchmarks for the evaluation of similarity measures: the multilingual benchmark and the image–caption benchmark.

- We show that ContraSim outperforms existing similarity measures in all benchmarks, and maintains a high accuracy even when faced with more challenging examples.

## 2 RELATED WORK

Comparing different models allows one to analyze how different aspects like network architecture, training set, and model size affect the model's learned representations. For instance, Kornblith et al. (2019) showed that adding too many layers to a convolutional neural network, trained for image classification, hurts its performance. Using CKA, they found that more than half of the network's layers are very similar to the last. They further found that two models trained on different image datasets (CIFAR-10 and CIFAR-100, Krizhevsky et al. 2009) learn representations that are similar in the shallow layers. Similar findings were noted for language models by Wu et al. (2020). The latter also evaluated the effect of fine-tuning on language models, and found that the top layers are most affected by fine-tuning.

Investigating the effect of layer width, Kornblith et al. (2019) and Morcos et al. (2018) found that increasing the model's layer width results in more similar representations between models, and that networks are generally more similar to networks with the same layer width than to networks with a relatively larger width. Raghu et al. (2017) provided an interpretation of the learning process by comparing the similarity of representations at some layer during the training process compared to the final representations. They found that networks converge from bottom to top, i.e., layers closer to the input converge to their final representation faster than deeper layers. Based on that insight, they proposed frozen training, where they successively freeze lower layers during training, updating only the deeper layers. They found that frozen training leads to classifiers with a higher generalization. Cianfarani et al. (2022) used similarity measures to analyze the effect of adversarial training on deep neural networks trained for image classification. Using CKA, they compared representations of adversarially trained neural networks with representations of regularly trained and discovered that adversarial examples have little effect on early layers. They further found that deeper layers overfit during adversarial training. Moreover, they found high similarity between representations of adversarial images generated with different threat model.

All prior work computes similarity only between examples that are similar, using functional closed-form measures. In contrast, we utilize both positive and negative samples in a learnable similarity measure, which allows adaptation to specific tasks.

## 3 PROBLEM SETUP

Let $\mathbb{X} = \{(x_1^{(i)}, x_2^{(i)})\}_{i=1}^N$ denote a set of $N$ examples, and $\mathbb{A} = \{(\boldsymbol{a}_1^{(i)}, \boldsymbol{a}_2^{(i)})\}_{i=1}^N$ the set of representations generated for the examples in $\mathbb{X}$. A representation is a high-dimensional vector of neuron activations. Representations may be created by the same or different models, by different layers of

the same model, etc. For instance, $x_1^{(i)}$ and $x_2^{(i)}$ may be the same input, with $\boldsymbol{a}_1^{(i)}$ and $\boldsymbol{a}_2^{(i)}$ representations of that input in different layers. Alternatively, $x_1^{(i)}$ can be an image and $x_2^{(i)}$ its caption, with $\boldsymbol{a}_1^{(i)}$ and $\boldsymbol{a}_2^{(i)}$ their representations from a vision model and a language model, respectively.

Our goal is to obtain a scalar similarity score, which represents the similarity between the two sets of representations, $\boldsymbol{a}_1^{(i)}$ and $\boldsymbol{a}_2^{(i)}$, and ranges from 0 (no similarity) to 1 (identical representations). That is, we define $\boldsymbol{X}_1 \in \mathbb{R}^{N \times p_1}$ as a matrix of $p_1$ activations of $N$ data points, and $\boldsymbol{X}_2 \in \mathbb{R}^{N \times p_2}$ as another matrix of $p_2$ activations of $N$ data points. We seek a similarity measure, $s(\boldsymbol{X}_1, \boldsymbol{X}_2)$.

## 4 CONTRASIM

In this section we introduce ContraSim, a similarity index for measuring the similarity of neural network representations. Our method uses a trainable encoder, which first maps representations to a new space and then measures the similarity of the projected representations. Formally, let $e_\theta$ denote an encoder with trainable parameters $\theta$, and assume two representations $\boldsymbol{a}_1$ and $\boldsymbol{a}_2$. In order to obtain a similarity score between 0 and 1, we first apply L2 normalization to the encoder outputs: $\boldsymbol{z}_1 = e_\theta(\boldsymbol{a}_1)/\|e_\theta(\boldsymbol{a}_1)\|$ (and similarly for $\boldsymbol{a}_2$). Then their similarity is calculated as:

$$s(\boldsymbol{z}_1, \boldsymbol{z}_2) \tag{1}$$

where $s$ is a simple closed-form similarity measure for two vectors. Throughout this work we use dot product for $s$.

For efficiency reasons, we calculate the similarity between batches of the normalized encoder representations, dividing by the batch size $n$:

$$\frac{1}{n} \sum_{i=1}^{n} \left( \boldsymbol{z}_1^i \cdot \boldsymbol{z}_2^i \right) \tag{2}$$

**Training.** None of the current similarity measures uses negative examples to estimate the similarity of a given pair. Using two examples, their output is usually a scalar that represents the similarity between them, without leveraging data from other examples. However, based on knowledge from other examples, we can construct a better similarity index. In particular, for a given example $x^{(i)} \in \mathbb{X}$ with its representation $\boldsymbol{a}_i$ we construct a set of *positive* examples indices, $P(i) = \{p_1, ..., p_q\}$, and a set of *negative* examples indices, $N(i) = \{n_1, ..., n_t\}$. The choice of these sets is task-specific and allows one to add inductive bias to the training process.

We train the encoder to maximize the similarity of $\boldsymbol{a}_i$ with all the positive examples, while at the same time making it dis-similar from the negative examples. We leverage ideas from contrastive learning (Chen et al., 2020; He et al., 2020), and minimize the following objective:

$$\mathcal{L} = \sum_{i \in I} \frac{-1}{|P(i)|} \log \frac{\sum_{p \in P(i)} \exp(\boldsymbol{a}_i \cdot \boldsymbol{a}_p / \tau)}{\sum_{n \in N(i)} \exp(\boldsymbol{a}_i \cdot \boldsymbol{a}_n / \tau)} \tag{3}$$

with scalar temperature parameter $\tau > 0$. Here $\boldsymbol{a}_p$ and $\boldsymbol{a}_n$ are representations from the positive and negative groups, respectively.

Our work uses negative examples and a trainable encoder for constructing a similarity measure. We evaluate these aspects in the experimental section, and show that using negative examples is an important aspect of our method. We show that the combination of the two leads to a similarity measure with superior results over current similarity measures.

## 5 SIMILARITY MEASURE EVALUATION

We use three benchmarks to evaluate similarity measures: the known layer prediction benchmark, and two new benchmarks we design: the multilingual benchmark and the image–caption benchmark. We further propose a strengthened version of the last two using the FAISS software.

Figure 1: Layer prediction benchmark. Given two models differing only in weight initialization, A and B, for each layer in the first model, among all layers of the second model, a good similarity measure is expected to assign the highest similarity for the architecturally-corresponding layer.

## 5.1 LAYER PREDICTION BENCHMARK

Proposed by Kornblith et al. (2019), a basic and intuitive benchmark is to assess the invariance of a similarity measure against changes to the random seed. Given two models, with the same architecture and training configuration (i.e., same training set, same hyperparameters, etc.), differing only in their weight initializations, for each layer in the first model, among all layers of the second model, one can expect that a good similarity measure assigns the highest similarity for the architecturally-corresponding layer. Formally, let $f$ and $g$ be two models with $k$ layers, and define as $f_i$ and $g_i$ as the $i^{th}$ layer of $f$ and $g$ models, respectively. After calculating the similarity of $f_i$ with each layer of $g$ ($g_1, \ldots, g_k$), the pair with the highest similarity is expected to be $(f_i, g_i)$. The benchmark counts the number of layers for which this pair was indeed assigned the highest similarity, and divides by the total number of pairs. An illustration is found in Figure 1.

The intuition behind this benchmark is that each layer captures different information about the input data. For example Jawahar et al. (2019) showed that different layers of the BERT model (Devlin et al., 2018) capture different semantic information. Kornblith et al. (2019) showed, using CKA similarity, that indeed there exists a correspondence between layers of models trained with different seeds. Thus, although trained from different seeds, the same layers are expected to capture the same information and therefore have a high similarity.

## 5.2 MULTILINGUAL BENCHMARK

Multilingual models, such as Multilingual-BERT (Devlin et al., 2018), learn to represent texts in different languages in the same representation space. Interestingly, these models show cross-lingual zero-shot transferability, where a model is fine-tuned in one language and evaluated in a different language (Pires et al., 2019). Muller et al. (2021) analyzed this transferability and found that lower layers of the Multilingual-BERT align the representations between sentences in different languages.

Since multilingual models share similarities between representations of different languages, we expect that a good similarity measure should assign a high similarity to two representations of a sentence in two different languages. In other words, we expect similarity measures to be invariant to the sentence source language. Consider a multilingual model $f$ and dataset $\mathbb{X}$, where each entry in it consists of the same sentence in different languages. Let $(x_1^{(i)}, x_2^{(i)}) \in \mathbb{X}$ be a sentence written in two languages – language A and language B. The similarity between $f(x_1^{(i)})$ and $f(x_2^{(i)})$ should be higher than the similarity between $f(x_1^{(i)})$ and the representation of a sentence in language B randomly chosen from $\mathbb{X}$, i.e., $f(x_2^{(j)})$, where $(x_1^{(j)}, x_2^{(j)}) \in \mathbb{X}$ is a randomly chosen example from $\mathbb{X}$. The benchmark calculates the fraction of cases for which the correct translation was assigned the highest similarity. An illustration is found in Figure 2.

Additionally, we suggest a strengthened version of the multilingual benchmark, using FAISS, a library for efficient similarity search. Instead of sampling random sentences in language B, we use FAISS to find the pair $(x_1^{(j)}, x_2^{(j)}) \in \mathbb{X}$, where $x_2^{(j)} \neq x_2^{(i)}$, with the representation $f(x_2^{(j)})$ that is most similar to $f(x_2^{(i)})$, out of a large set of vectors pre-indexed by FAISS. This leads to a more challenging scenario, as the similarity between $x_1^{(i)}$ and FAISS sampled $x_2^{(j)}$ is expected to be higher than the similarity between $x_1^{(i)}$ and randomly chosen $x_2^{(j)}$, increasing the difficulty of the similarity measure to correctly identify the pair $(x_1^{(i)}, x_2^{(i)})$ as the highest-similarity pair.

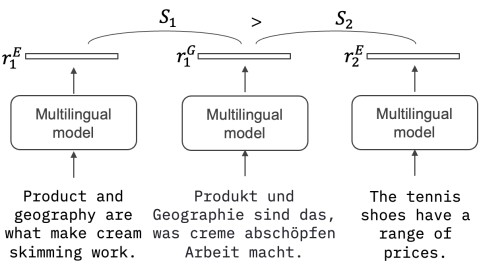 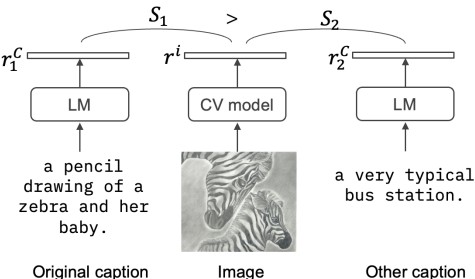

Figure 2: The multilingual benchmark. $r_1^E$ and $r_1^G$ denote the representations of the same sentence in different languages, and $S_1$ is their similarity. $r_2^E$ represents the random sentence representation, and $S_2$ is the similarity between it and $r_1^G$. We expect $S_1$ to be higher than $S_2$.

Figure 3: The image–caption benchmark. $r_1^C$ and $r^i$ denote the representations of the caption and the image pair, respectively, and $S_1$ is their similarity. $r_2^C$ denotes the random caption representation, and $S_2$ is the similarity between it and $r^i$. We expect $S_1$ to be higher than $S_2$.

### 5.3 IMAGE-CAPTION BENCHMARK

Let $\mathbb{X}$ be a dataset of images and their textual descriptions (captions), $f$ be a computer vision model and $g$ a language model. Given a pair of an image and its caption, $(m^{(i)}, c^{(i)}) \in \mathbb{X}$, a good similarity measure is expected to assign a high similarity to their representations – $f(m^{(i)}), g(c^{(i)})$. In particular, this similarity should be higher than that of the pair of the same image representation $f(m^{(i)})$ and some random caption's representation $g(c^{(j)})$, where $c^{(j)}$ is a randomly chosen caption from dataset $\mathbb{X}$. The intuition behind this benchmark is that an image and its caption represent the same scene in a different way. Thus, their representations should have a higher similarity than that of the same image and some random caption. An illustration is found in Figure 3.

As in the multilingual benchmark, we also propose a strengthened variant for the image–caption benchmark using FAISS. Rather than sampling a random caption $c^{(j)}$, we use FAISS to find the pair $(m^{(j)}, c^{(j)}) \in \mathbb{X}$, where $c^{(j)} \neq c^{(i)}$, with the representation $g(c^{(j)})$ that is most similar to $g(c^{(i)})$.

## 6 EXPERIMENTS

**Baselines and Ablations.** We compare ContraSim with the following standard baselines.

- **Centered Kernel Alignment (CKA)**: Proposed by Kornblith et al. (2019), CKA computes a kernel matrix for each matrix representation input, and defines the scalar similarity index as the two kernel matrices' alignment. We use a linear kernel for CKA evaluation, as the original paper reveals similar results for both linear and RBF kernels. CKA is our main point of comparison due to its success in prior work and wide applicability.

- **PWCCA**: Proposed by Morcos et al. (2018), PWCCA is an extension of Canonical Correlation Analysis (CCA). Given two matrices, CCA finds bases for those matrices, such that after projecting them to those bases the correlation between the projected matrices is maximized. While in CCA the scalar similarity index is computed as the mean correlation coefficient, in PWCCA that mean is weighted by the importance each canonical correlation has on the representation.[1]

See Appendix A.4 for more details on each method. Appendix A.5 report additional baseline results.

In addition, we report the results of two new similarity measures, which use an encoder to map representations to the space where similarity is measured. However, in both methods we train $f_\theta$ to only maximize the similarity between positive pairs:

$$\mathcal{L}_{max} = -s(z_1, z_2) \tag{4}$$

---

[1]PWCCA requires the number of examples to be larger than the feature vector dimension, which is not possible to achieve in all benchmarks. Therefore, we compare with PWCCA in a subset of our experiments.

where $z_1$ and $z_2$ are representations whose similarity we wish to maximize, where for a batch of size $n$ similarity is measured as: $\frac{1}{n}\sum_{i=1}^{n} s\left(z_1^i, z_2^i\right)$. We experiment with two functions for $s$— dot-product and CKA—and accordingly name these similarity measures DeepDot and DeepCKA. These methods provide a point of comparison where the similarity measure is trained, but *without negative examples*, to examine whether contrastive learning is crucial to our method's success.

**Encoders details.** In all experiments, the encoder $f_\theta$ is a two-layer multi-layered perceptron with hidden layer dimensions of $512$ and $256$, and output dimension of $128$. We trained the encoder for 50 epochs for the layer prediction and 30 epochs for the multilingual and image–caption benchmarks. We used the Adam optimizer (Kingma & Ba, 2014) with a learning rate of $0.001$ and a batch size of $1024$ representations. We used $\tau = 0.07$ for ContraSim training.

## 6.1 LAYER PREDICTION BENCHMARK

### 6.1.1 SETUP

Recall that this benchmark evaluates whether a certain layer from one model is deemed most similar to its architecturally-corresponding layer from another model, where the two models differ only in their weight initialization. We repeat this process for all layers and 5 different model pairs, and report average accuracy. We experiment with both language and vision setups.

**Models.** For language experiments, we use the recent MultiBERTs (Sellam et al., 2021), a set of 25 BERT models, differing only in their initial random weights. For vision experiments, we pre-train 10 visual transformer (ViT) models (Dosovitskiy et al., 2020) on the ImageNet-1k dataset (Russakovsky et al., 2015). Then we fine-tune them on CIFAR-10 and CIFAR-100 datasets (Krizhevsky et al., 2009). Further training details are available in Appendix A.3.

**Datasets.** For language experiments, we experiment with word-level contextual representations generated on two English text datasets: the Penn Treebank (Marcus et al., 1993) and WikiText (Merity et al., 2016). For Penn TreeBank we generate 5005 and 10019 representations for the test and training sets, respectively; for WikiText we generate 5024/10023 test/training representations. For vision experiments, we experiment with representations generated on CIFAR-10 and CIFAR-100. For both we generate 5000 and 10000 test and training representations, respectively.

**Positive and Negative sets.** Given a batch of representations of some model $i$ at layer $j$, we define its positive set as the representations at the same layer $j$ of all models that differ from $i$. The negative set is all representations from layers that differ from $j$ (including from model $i$).

### 6.1.2 RESULTS

The results are shown in Table 1. In both language and vision evaluations, CKA achieves better results than PWCCA, consistent with the findings by (Ding et al., 2021). DeepDot and DeepCKA perform poorly, with much lower results than PWCCA and CKA, revealing that maximizing the similarity is not satisfactory for similarity measure purposes. Our method, ContraSim, achieves excellent results. When trained on one dataset's training set and evaluated on the same dataset's test set, ContraSim achieves perfect accuracy under this benchmark, with a large margin over CKA results. This holds for both language and vision cases.

Even when trained on one dataset and evaluated over another dataset, ContraSim surpasses other similarity measures results, showing the transferability of the learned encoder projection between datasets. This is true both when transferring across domains (in text, between news texts from the Penn Treebank and Wikipedia texts), and when transferring across classification tasks (in images, between the 10-label CIFAR-10 and the 100-label CIFAR-100).

## 6.2 MULTILINGUAL BENCHMARK

### 6.2.1 SETUP

This benchmark assesses whether a similarity measure assigns a high similarity to multilingual representations of the same sentence in different languages. Given a batch of (representations of) sen-

Table 1: Layer prediction benchmark accuracy results for language and vision cases. For encoder-based methods we report mean and std over 5 random initializations. For ContraSim, we experiment with training with different datasets (rows) and evaluating on same or different datasets (columns).

|  | Language | |  | Vision | |
| --- | --- | --- | --- | --- | --- |
|  | Penn TreeBank | WikiText |  | CIFAR-10 | CIFAR-100 |
| PWCCA | 38.33 | 55.00 | PWCCA | 47.27 | 45.45 |
| CKA | 71.66 | 76.66 | CKA | 78.18 | 74.54 |
| DeepDot | $15.55 \pm 1.69$ | $14.00 \pm 2.26$ | DeepDot | $14.90 \pm 1.78$ | $14.18 \pm 2.67$ |
| DeepCKA | $16.66 \pm 3.16$ | $19.66 \pm 1.63$ | DeepCKA | $17.09 \pm 2.95$ | $13.09 \pm 4.20$ |
| ContraSim | | | ContraSim | | |
| Penn | $100 \pm 0$ | $85.45 \pm 1.62$ | CIFAR-10 | $100 \pm 0$ | $90.54 \pm 2.90$ |
| Wiki | $94.00 \pm 4.66$ | $100 \pm 0$ | CIFAR-100 | $85.81 \pm 5.68$ | $100 \pm 0$ |

tences $b^{(i)}$ in language $L_i$ and their translations $b^{(j)}$ in language $L_j$, we compute the similarity between $b^{(i)}$ and $b^{(j)}$, and the similarities between $b^{(i)}$ and 10 randomly chosen batches of representations in language $L_j$. If $b^{(i)}$ is more similar to $b^{(j)}$ than it is to all other batches, then we mark success. (Alternatively, in a more challenging scenario, we use FAISS to find for each representation in each layer the 10 most similar representations in that layer.) We repeat this process separately for representations from different layers of a multilingual model, over many sentences and multiple language pairs, and report average accuracy per layer.[2] Appendix A.1 gives more details.

**Model and Data.** We use two multilingual models: multilingual BERT (Devlin et al., 2018)[3] and XLM-R (Conneau et al., 2019). We use the XNLI dataset (Conneau et al., 2018), which has natural language inference examples, parallel in multiple languages. Each example in our dataset is a sentence taken from either the premise or hypothesis sets. We experiment with 5 typologically-different languages: English, Arabic, Chinese, Russian, and Turkish. We created sentence-level representations, with 5000 test 10000 training representations. As a sentence representation, we experiment with [CLS] token representations and with mean pooling of token representations, since Del & Fishel (2021) noted a difference in similarity in these two cases. We report results with [CLS] representations in the main paper and mean pooling results in Appendix A.1; the trends are similar.

**Positive and Negative sets.** Given a pair of languages and a batch of representations at some layer, for each representation we define its positive pair as the representation of the sentence in the different language, and its negative set as all other representations in the batch.

### 6.2.2 RESULTS

Results with multilingual BERT representations in Table 2 show the effectiveness of our method. (trends with XLM-R are consistent; Appendix A.1.3). Under random sampling evaluation (left block), ContraSim shows superior results over other similarity measures, although always being evaluated on language pairs it has not seen at training time. Using a FAISS-based sampler (right block) further extends the gaps. While CKA results dropped by $\approx 45\%$, DeepCKA dropped by $\approx 51\%$, and DeepDot dropped by $\approx 40\%$, ContraSim was much less affected by FAISS sampling ($\approx 17\%$ drop on average and practically no drop in most layers). This demonstrates the high separability between examples of ContraSim, enabling it to distinguish even very similar examples and assign a higher similarity to the correct pair. For all other methods, mid-layers have the highest accuracy, whereas for our method almost all layers are near $100\%$ accuracy, except for the first 3 or 4 layers.

Evaluation results reveals interesting insights that were not found using previous similarity measures. In FAISS results, we see that there is a much greater difference in accuracy between shallow and deep layers in ContraSim compared to previous similarity measures. This means that using previous similarity measures we might infer that there is no difference in the ability to detect the

---

[2]For deep similarity measures (DeepCKA, DeepDot, and ContraSim), upon training the encoder on examples from a pair of languages, $(L_r, L_q), r \neq q$, we evaluate it over all other distinct pairs of languages.

[3]https://huggingface.co/bert-base-multilingual-cased

Table 2: Multilingual benchmark accuracy results. With random sampling (left block), ContraSim outperforms other similarity measures. Using FAISS (right block) further extends the gaps.

| | Random | | | | FAISS | | | |
|---|---|---|---|---|---|---|---|---|
| Layer | CKA | DeepCKA | DeepDot | ContraSim | CKA | DeepCKA | DeepDot | ContraSim |
| 1 | $71.7 \pm 5.3$ | $82.0 \pm 6.4$ | $63.3 \pm 10.4$ | $95.5 \pm 5.4$ | $20.1 \pm 4.0$ | $10.7 \pm 2.6$ | $29.9 \pm 8.7$ | $36.0 \pm 10.7$ |
| 2 | $78.7 \pm 4.4$ | $86.4 \pm 4.1$ | $68.5 \pm 9.9$ | $95.0 \pm 7.2$ | $27.2 \pm 5.5$ | $12.3 \pm 2.9$ | $46.9 \pm 9.8$ | $33.0 \pm 14.8$ |
| 3 | $86.8 \pm 3.0$ | $87.1 \pm 3.2$ | $70.4 \pm 9.7$ | $96.4 \pm 6.7$ | $41.9 \pm 8.7$ | $17.6 \pm 4.2$ | $51.5 \pm 10.3$ | $45.4 \pm 20.5$ |
| 4 | $92.6 \pm 1.4$ | $91.5 \pm 2.4$ | $95.4 \pm 3.4$ | $99.9 \pm 0.2$ | $33.4 \pm 7.0$ | $15.2 \pm 3.7$ | $52.2 \pm 8.6$ | $72.4 \pm 9.8$ |
| 5 | $88.3 \pm 3.2$ | $83.5 \pm 5.2$ | $94.7 \pm 4.8$ | $99.9 \pm 0$ | $49.3 \pm 4.3$ | $36.9 \pm 6.3$ | $42.4 \pm 12.9$ | $99.1. \pm 0.8$ |
| 6 | $88.6 \pm 3.4$ | $86.4 \pm 5.2$ | $92.5 \pm 5.4$ | $100 \pm 0$ | $51.4 \pm 5.5$ | $39.9 \pm 7.2$ | $42.1 \pm 12.3$ | $99.5. \pm 0.4$ |
| 7 | $88.8 \pm 3.7$ | $86.9 \pm 5.0$ | $92.6 \pm 5.0$ | $100 \pm 0$ | $53.0 \pm 5.8$ | $41.1 \pm 7.7$ | $45.7 \pm 11.7$ | $99.6. \pm 0.3$ |
| 8 | $89.3 \pm 3.6$ | $85.2 \pm 5.7$ | $91.4 \pm 7.0$ | $100 \pm 0$ | $56.1 \pm 5.8$ | $45.0 \pm 8.7$ | $43.8 \pm 13.4$ | $99.7. \pm 0.3$ |
| 9 | $88.1 \pm 3.8$ | $82.4 \pm 5.6$ | $89.1 \pm 9.5$ | $100 \pm 0$ | $53.3 \pm 4.9$ | $42.7 \pm 8.5$ | $39.2 \pm 12.9$ | $99.6. \pm 0.3$ |
| 10 | $87.0 \pm 3.5$ | $80.3 \pm 5.9$ | $85.3 \pm 10.3$ | $100 \pm 0$ | $51.5 \pm 5.3$ | $42.4 \pm 7.8$ | $34.3 \pm 12.2$ | $99.5. \pm 0.4$ |
| 11 | $86.7 \pm 4.2$ | $76.6 \pm 6.4$ | $79.7 \pm 13.9$ | $99.9 \pm 0$ | $52.4 \pm 5.3$ | $43.3 \pm 8.5$ | $31.4 \pm 12.8$ | $99.3. \pm 0.5$ |
| 12 | $86.4 \pm 3.4$ | $63.8 \pm 7.9$ | $64.3 \pm 19.7$ | $99.9 \pm 0$ | $52.8 \pm 4.5$ | $32.3 \pm 8.7$ | $26.1 \pm 21.9$ | $98.9. \pm 0.8$ |

correct pair across different layers. However, ContraSim shows that the difference in the ability to detect the correct pair dramatically changes from shallow to deep layers.

To further analyze this, we compare the original multilingual representations from the last layer with their projections by ContraSim's trained encoder. Figure 5 from Appendix A.1.2 shows UMAP (McInnes et al., 2018) projections of representations of 5 sentences in English and 5 sentences in Arabic, before and after ContraSim encoding. The ContraSim encoder was trained on Arabic and English languages. The original representations are organized according to the source language (by shape), whereas ContraSim projects translations of the same sentence close to each other (clustered by color).

### 6.3 IMAGE–CAPTION BENCHMARK

#### 6.3.1 SETUP

Given a test set $\mathbb{X}$, consisting of pairs of an image representation generated by a computer vision model and its caption representation from a language model, we split $\mathbb{X}$ to batches of size 64. For each batch, we compute the similarity between the image representations and their corresponding caption representations. We then sample 10 different caption batches, either randomly or using FAISS (as before), and compute the similarity between the image representation and each random/FAISS-retrieved caption representation. If the highest similarity is between the image representation and the original caption representation, we mark a success. For trainable similarity measures, we train with 5 different random seeds and average the results.

**Models and Data.** We use two vision models for image representations: ViT and ConvNext (Liu et al., 2022); and two language models for text representations: BERT and GPT2 (Radford et al., 2019). We use the Conceptual Captions dataset (Sharma et al., 2018), which has ≈3.3M pairs of images and English captions. We use 5000 and 10000 pairs as test and training sets, respectively.

**Positive and Negative sets.** Given a batch of image representations with their corresponding caption representations, for each image representation we define as a positive set its corresponding caption representation, and as a negative set all other representations in the batch.

#### 6.3.2 RESULTS

Figure 4 demonstrates the strength of ContraSim. Under random sampling (green boxes), Deep-CKA achieves comparable results to ContraSim, while DeepDot and CKA achieve lower results. However, using FAISS (red boxes) causes a big decrease in DeepCKA accuracy, while ContraSim maintains high accuracy.

Another interesting result is that in 3 of 4 pairs we tested, CKA accuracy using FAISS for sampling was higher than using random sampling. This contradicts the intuition that using similar examples at the sampling stage should make it difficult for similarity measures to distinguish between examples. This might indicate that CKA suffers from stability issues.

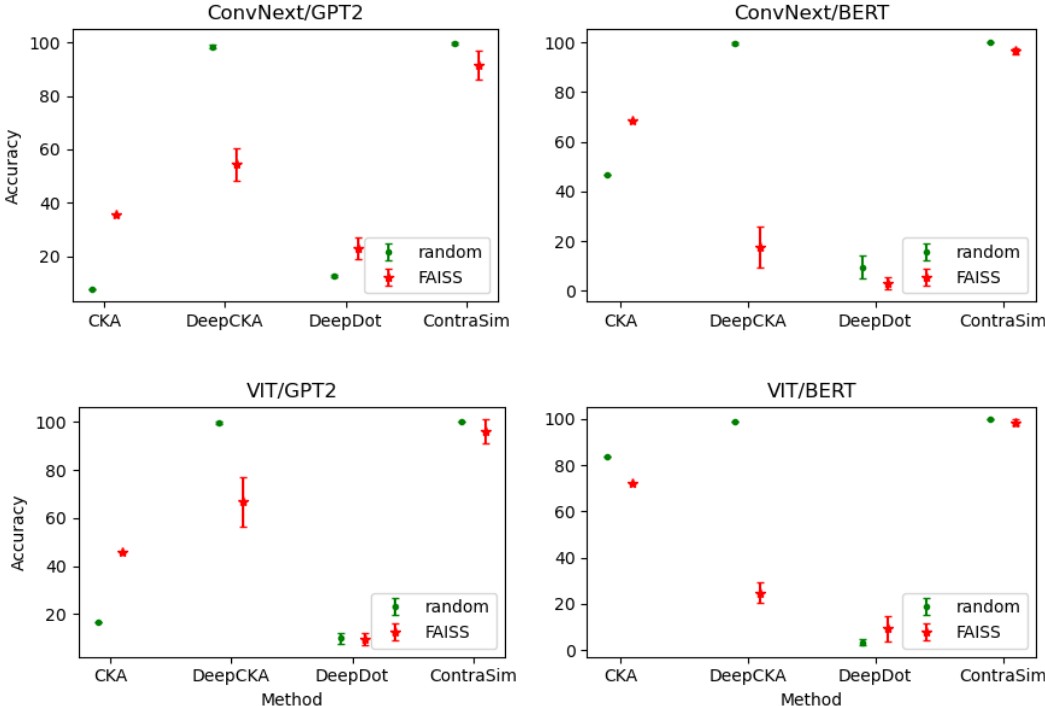

Figure 4: Image–caption benchmark results for 4 different model pairs. ContraSim works best, and is the only measure robust to FAISS sampling.

Looking at CKA results we might observe that BERT representations are more similar to computer vision representations rather than GPT2 model representations. That is because using CKA we see that the pairs with BERT model achieve higher accuracy compared to pairs with GPT2 model. Higher accuracy means that it is easier to detect the matching pair, which means that the representations are more similar. However, ContraSim achieves high accuracy in both BERT pairs and GPT2 pairs, which means that both models share the same similarity to computer vision models, in contrast to what we may infer from previous similarity measures.

Finally, we report results with the multi-modal CLIP model (Radford et al., 2021) in Appendix 5. Because the model was pre-trained with contrastive learning, simple dot-product similarity works very well, so there is no need to learn a similarity measure in this case.

## 7 CONCLUSION

We proposed a new similarity measure, ContraSim, based on ideas from contrastive learning. By defining the positive and negative sets we learn an encoder that maps representation to a space where similarity is measured. Our method outperformed other similarity measures under the common layer prediction benchmark, and two new benchmarks we proposed: the multilingual benchmark and the image–caption benchmark. It particularly shines in strengthened versions of said benchmarks, where random sampling is replaced with finding the most similar examples using FAISS.

Our new similarity measure benchmarks can facilitate work on a similarity-based analysis of deep neural networks. The multilingual benchmark is useful for work in multilingual language models, while the image–caption benchmark may help work in multi-modal settings. A drawback of our method is that it can only compare representations of the same dimensionality. This can be addressed by using a dimensionality reduction to a shared space. We leave the research of a more sophisticated method for future work. Moreover, compared to existing methods, ContraSim needs access to a training set for the encoder training procedure. Compared with closed-form similarity measures, train time is another trade-off. In addition, since our method learns a parameterized measure, it may help train models with similarity objectives. We also leave that for future work. Finally, considering ContraSim's superiority in all evaluations, it will better fit for interpretability of neural networks.

ETHICS STATEMENT

Our work adds to the body of literature on the interpretability of neural networks and may mitigate their opacity. We do not foresee major risks associated with this work. However, a malicious actor could train ContraSim adversarially, assign poor similarity estimates, and lead to false analyses.

REPRODUCIBILITY STATEMENT

All the evaluations conducted throughout this paper are fully reproducible. ContraSim training scheme and architecture were described in Sections 4 and 6. Detailed information regarding each evaluation is also provided in the Section 6, with further details in Appendix A. Code to reproduce all the paper results will be made publicly available upon publication.

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

Table 3: Multilingual benchmark results with mean pooling.

| | | Random | | | FAISS | | | |
|---|---|---|---|---|---|---|---|---|
| Layer | CKA | DeepCKA | DeepDot | ContraSim | CKA | DeepCKA | DeepDot | ContraSim |
| 1 | $87.7 \pm 6.9$ | $86.3 \pm 9.7$ | $43.4 \pm 17.7$ | $98.7 \pm 2.2$ | $67.6 \pm 14.3$ | $54.1 \pm 10.9$ | $41.7 \pm 19.1$ | $94.2 \pm 10.7$ |
| 2 | $89.0 \pm 6.3$ | $88.7 \pm 7.0$ | $51.5 \pm 20.0$ | $99.5 \pm 0.8$ | $68.2 \pm 13.2$ | $49.0 \pm 8.3$ | $38.9 \pm 17.2$ | $96.6 \pm 14.8$ |
| 3 | $91.8 \pm 4.4$ | $90.7 \pm 6.0$ | $63.3 \pm 20.8$ | $99.9 \pm 0.1$ | $72.2 \pm 11.5$ | $55.4 \pm 8.1$ | $44.8 \pm 16.6$ | $98.8 \pm 20.5$ |
| 4 | $93.7 \pm 3.3$ | $91.3 \pm 5.0$ | $73.1 \pm 19.4$ | $99.9 \pm 0.0$ | $74.3 \pm 10.0$ | $55.1 \pm 8.0$ | $45.7 \pm 16.5$ | $99.5 \pm 7.1$ |
| 5 | $95.3 \pm 3.0$ | $92.1 \pm 4.0$ | $83.9 \pm 15.6$ | $99.9 \pm 0.0$ | $78.2 \pm 8.2$ | $56.7 \pm 8.1$ | $53.2 \pm 17.5$ | $99.8 \pm 4.4$ |
| 6 | $95.9 \pm 2.4$ | $91.8 \pm 3.9$ | $91.2 \pm 10.6$ | $100 \pm 0$ | $77.6 \pm 7.9$ | $54.2 \pm 8.1$ | $60.1 \pm 18.2$ | $99.8 \pm 1.7$ |
| 7 | $95.4 \pm 2.5$ | $90.6 \pm 4.1$ | $93.1 \pm 9.2$ | $100 \pm 0$ | $77.9 \pm 7.8$ | $53.3 \pm 7.2$ | $63.5 \pm 18.5$ | $99.9 \pm 0.7$ |
| 8 | $94.8 \pm 3.2$ | $89.7 \pm 4.3$ | $90.3 \pm 12.0$ | $100 \pm 0$ | $76.7 \pm 8.1$ | $52.4 \pm 7.4$ | $61.0 \pm 19.8$ | $99.9 \pm 0.3$ |
| 9 | $94.0 \pm 3.4$ | $88.5 \pm 5.0$ | $86.4 \pm 15.1$ | $100 \pm 0$ | $73.9 \pm 8.8$ | $51.4 \pm 7.8$ | $55.5 \pm 20.0$ | $99.9 \pm 0.1$ |
| 10 | $92.6 \pm 4.2$ | $85.6 \pm 5.9$ | $80.7 \pm 18.8$ | $100 \pm 0$ | $72.2 \pm 8.4$ | $49.3 \pm 8.4$ | $49.2 \pm 20.6$ | $99.9 \pm 0.1$ |
| 11 | $91.1 \pm 5.1$ | $81.0 \pm 6.5$ | $72.2 \pm 23.7$ | $99.9 \pm 0$ | $70.6 \pm 10.1$ | $48.8 \pm 9.1$ | $43.2 \pm 20.7$ | $99.8 \pm 0.1$ |
| 12 | $90.8 \pm 5.8$ | $71.3 \pm 7.6$ | $71.0 \pm 21.0$ | $99.9 \pm 0$ | $72.7 \pm 11.3$ | $40.3 \pm 8.7$ | $42.7 \pm 17.0$ | $99.4 \pm 0.1$ |

# A APPENDIX

## A.1 MULTILINGUAL BENCHMARK

### A.1.1 EVALUATION PARAMETERS

We split the test set, $\mathbb{X}$, into equally sized batches of size 8, $\{b^{(1)}, b^{(2)}, ..., b^{(n)}\}$, where each batch consists of multilingual BERT representations of the same sentence in 5 different languages: $L = \{L_1, ..., L_5\}$. Given a pair of different languages, $(L_i, L_j), i \neq j$, and a batch of representations, $b$, we consider the representation of those languages in the batch, $(b[i], b[j])$, and compute the similarity between $b[i]$ and $b[j]$ as $s_0 \equiv s(b[i], b[j])$. We also compute the similarity between $b[i]$ and 10 randomly chosen batches (or, 10 batches chosen using FAISS) of representations in language $L_j$ as $\{s_t \equiv s(b[i], b_t[j])\}_{t=1}^{10}$. If $\arg\max_t \{s_t\}_{t=0}^{10} = 0$, we count it as a correct prediction. Each layer's accuracy is defined as the number of successful predictions over the number of batches, $n$. We average results over all possible pairs of different languages.

### A.1.2 CONTRASIM PROJECTIONS

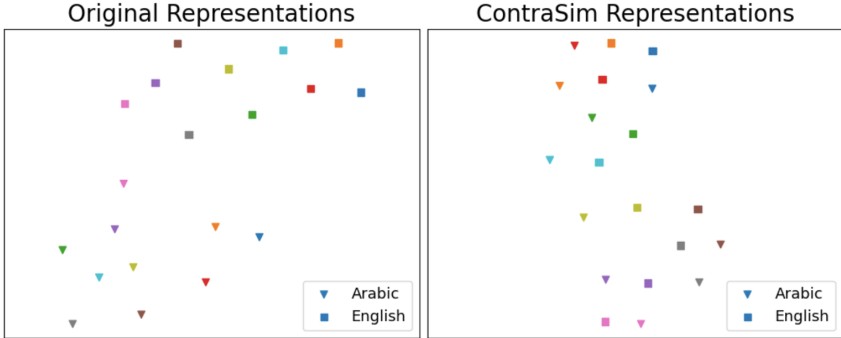

Figure 5: Original representations (left) are clustered by the source language (by shape). ContraSim (right) projects representations of the same sentence in different languages close by (by color).

### A.1.3 FURTHER EVALUATIONS

In addition to using the [CLS] token representation as a sentence representation, we also evaluated the multilingual benchmark using mean pooling sentence representation. We used the same evaluation process as described in Section 6.2. The results, summarized in Table 3, are consistent with the results in the main paper (Table 2). Under random sampling, ContraSim outperforms all other similarity measures. Using FAISS causes a big degradation in all other methods' accuracy, while ContraSim maintains a high accuracy across all layers.

In addition, we evaluated the multilingual benchmark with another multilingual model – the XLM-R (Conneau et al., 2019) model. Results, summarized in Table 4, show a similar pattern to Tables 2 and 3, with ContraSim achieving the highest accuracies across all layers, in both random sampling and FAISS sampling scenarios.

Table 4: Multilingual benchmark results on XLM-R model.

| Layer | Random | | | | FAISS | | | |
|---|---|---|---|---|---|---|---|---|
| | CKA | DeepCKA | DeepDot | ContraSim | CKA | DeepCKA | DeepDot | ContraSim |
| 1 | $89.7 \pm. 4.6$ | $88.2 \pm 3.1$ | $45.0 \pm 22.1$ | $99.89 \pm 0.31$ | $60.6 \pm 11.4$ | $24.6 \pm 4.7$ | $24.8 \pm 12.4$ | $98.1 \pm 2.5$ |
| 2 | $90.6 \pm 4.1$ | $92.3 \pm 2.0$ | $63.1 \pm 23.4$ | $99.99 \pm 0.02$ | $57.1 \pm 11.4$ | $30.7 \pm 5.7$ | $31.1 \pm 13.5$ | $99.5 \pm 0.7$ |
| 3 | $92.8 \pm 3.1$ | $93.8 \pm 1.7$ | $79.5 \pm 18.9$ | $99.99 \pm 0$ | $55.5 \pm 10.1$ | $33.8 \pm 6.4$ | $39.3 \pm 15.5$ | $99.9 \pm 0.1$ |
| 4 | $94.7 \pm 2.6$ | $94.3 \pm 1.7$ | $91.4 \pm 11.7$ | $100 \pm 0$ | $62.3 \pm 9.5$ | $36.3 \pm 6.8$ | $55.1 \pm 16.2$ | $99.9 \pm 0$ |
| 5 | $95.9 \pm 2.1$ | $94.6 \pm 1.6$ | $94.0 \pm 10.0$ | $100 \pm 0$ | $66.2 \pm 8.4$ | $37.2 \pm 7.6$ | $64.2 \pm 16.2$ | $99.9 \pm 0$ |
| 6 | $95.9 \pm 2.1$ | $94.9 \pm 1.6$ | $94.6 \pm 8.9$ | $100 \pm 0$ | $66.7 \pm 8.3$ | $41.2 \pm 7.6$ | $66.0 \pm 17.5$ | $99.9 \pm 0$ |
| 7 | $96.6 \pm 2.0$ | $94.9 \pm 1.6$ | $94.9 \pm 8.5$ | $100 \pm 0$ | $71.7 \pm 8.5$ | $44.1 \pm 8.2$ | $68.8 \pm 17.5$ | $99.9 \pm 0$ |
| 8 | $96.1 \pm 2.1$ | $94.8 \pm 1.7$ | $94.0 \pm 9.3$ | $100 \pm 0$ | $68.1 \pm 8.3$ | $43.6 \pm 7.9$ | $65.3 \pm 18.0$ | $99.9 \pm 0$ |
| 9 | $94.8 \pm 2.2$ | $94.9 \pm 1.6$ | $93.3 \pm 9.1$ | $100 \pm 0$ | $58.5 \pm 8.7$ | $42.8 \pm 8.0$ | $61.7 \pm 18.6$ | $99.9 \pm 0$ |
| 10 | $93.9 \pm 2.3$ | $94.3 \pm 1.7$ | $92.7 \pm 10.7$ | $100 \pm 0$ | $46.3 \pm 8.2$ | $39.1 \pm 7.5$ | $56.6 \pm 18.9$ | $99.9 \pm 0$ |
| 11 | $92.0 \pm 2.8$ | $93.3 \pm 2.2$ | $92.7 \pm 10.9$ | $100 \pm 0$ | $35.5 \pm 7.0$ | $39.1 \pm 7.3$ | $57.2 \pm 18.3$ | $99.9 \pm 0$ |
| 12 | $80.7 \pm 4.7$ | $89.5 \pm 3.0$ | $81.6 \pm 13.8$ | $100 \pm 0$ | $26.5 \pm 5.8$ | $32.3 \pm 7.0$ | $34.7 \pm 15.1$ | $99.9 \pm 0$ |

## A.2 IMAGE–CAPTION

In addition to the four model pairs we evaluated in Figure 4, we assessed the multi-modal vision and language CLIP model (Radford et al., 2021), which was trained using contrastive learning on pairs of images and their captions. Results in Table 5 show interesting findings. Under random sampling, dot product, DeepCKA and ContraSim achieve perfect accuracy. However, using FAISS causes significant degradation in DeepCKA accuracy, and only a small degradation in dot product and ContraSim results, with equal accuracy for both. We attribute this high accuracy for simple dot product to the fact that CLIP training was done using contrastive learning, thus observing high separability between examples.

Table 5: Image–caption benchmark accuracy results using CLIP model

| | Random | FAISS |
|---|---|---|
| CKA | 93.67 | 25.31 |
| Dot Product | 100 | 98.73 |
| DeepCKA | 100 | 13.92 |
| DeepDot | 29.11 | 25.31 |
| ContraSim | 100 | 98.73 |

## A.3 VIT TRAINING DETAILS

We used the ViT-base (Dosovitskiy et al., 2020) architecture. We pretrained 10 models on the ImgaeNet-1K dataset (Deng et al., 2009), differing only in their weight initializations by using random seeds from 0 to 9. We used the AdamW optimizer (Kingma & Ba, 2014) with $lr = 0.001$, weight decay $= 1e - 3$, batch size $= 128$, and a cosine learning scheduler. We trained each model for 150 epochs and used the final checkpoint.

Then, we fine-tuned the pretrained models on CIFAR-10 and CIFAR-100 datasets (Krizhevsky et al., 2009). We used AdamW optimizer with $lr = 2e - 5$, weight decay $= 0.01$, batch size $= 10$, and a linear scheduler. For models fine-tuned on CIFAR-10, the average accuracy on the CIFAR-10 test set is 96.33%. For models fine-tuned on CIFAR-100, the average accuracy on the CIFAR-100 test set is 78.87%.

### A.4 Details of Prior Similarity Measures

**Canonical Correlation Analysis (CCA).** Given two matrices, CCA finds bases for those matrices, such that after projecting them to those bases the projected matrices' correlation is maximized. For $1 \leq i \leq p_1$, the $i^{\text{th}}$ canonical correlation coefficient $\rho_i$ is given by:

$$\rho_i = \max_{w_X^i, w_Y^i} \text{corr}(Xw_X^i, Yw_Y^i)$$
$$\text{s.t.} \quad \forall_{j<i} \ Xw_X^i \perp Xw_X^j \tag{5}$$
$$\forall_{j<i} \ Yw_Y^i \perp Yw_Y^j.$$

where $\text{corr}(X,Y) = \frac{\langle X,Y \rangle}{\|X\| \cdot \|Y\|}$. Given the vector of correlation coefficients $\text{corrs} = (\rho_1, ..., \rho_{p_1})$, the final scalar similarity index is computed as the mean correlation coefficient:

$$S_{\text{CCA}}(X,Y) = \overline{\rho}_{CCA} = \frac{\sum_{i=1}^{p_1} \rho_i}{p_1} \tag{6}$$

as previously used in (Raghu et al., 2017; Kornblith et al., 2019).

**Projection-Weighted CCA (PWCCA).** Morcos et al. (2018) proposed a different approach to transform the vector of correlation coefficients, corrs, into a scalar similarity index. Instead of defining the similarity as the mean correlation coefficient, PWCCA uses a weighted mean and the similarity is defined as:

$$S_{PW} = \frac{\sum_{i=1}^{p_1} \alpha_i \rho_i}{\sum_i \alpha_i} \qquad \alpha_i = \sum_j |\langle h_i, x_j \rangle| \tag{7}$$

where $x_j$ is the $j^{\text{th}}$ column of $X$, and $h_i = Xw_X^i$ is the vector observed upon projecting $X$ to the $i^{\text{th}}$ canonical direction. Code available at: `https://github.com/google/svcca`.

**Centered Kernel Alignment (CKA).** CKA, Proposed by Kornblith et al. (2019), suggests computing a kernel matrix for each matrix representation input, and defining the scalar similarity index as the two kernel matrices' alignment. For linear kernel, CKA is defined as:

$$S_{\text{CKA}} = \frac{\|Y^T X\|_F^2}{\|X^T X\|_F \|Y^T Y\|_F} \tag{8}$$

Code available at: `https://github.com/google-research/google-research/tree/master/representation_similarity`.

### A.5 Additional Similarity Measures

We evaluated three more similarity measures: singular vector CCA (SVCCA) Raghu et al. (2017), dot product and $L_2$ norm between the difference of the normalized representations (dubbed Norm in the paper). Given two batch representations, $X$ and $Y$, SVCCA performs CCA on the truncated singular value decomposition (SVD) of $X$ and $Y$.

For two representations, $x$ and $y$, we defined dis-similarity measure as:

$$Dis_{\text{Norm}}(x,y) = \|(x/\|x\| - y/\|y\|)\| \tag{9}$$

Since this is a dis-similarity measure, we defined the norm similarity measure as:

$$S_{\text{Norm}} = 1 - Dis_{\text{Norm}}(x,y) \tag{10}$$

For a batch of representations, we define batch similarity as the mean of pairwise norm similarity. In addition, we evaluated ContraSim with a different similarity measure than dot-product and replaced it with the norm similarity measure.

Table 6: Layer prediction benchmark with additional similarity measures.

|  | Penn TreeBank | WikiText |
|---|---|---|
| SVCCA | 46.66 | 56.66 |
| Dot product | 8.33 | 6.66 |
| Norm | 10.00 | 11.66 |
| ContraSim_norm |  |  |
| Penn | 100.00 | 90.00 |
| Wiki | 100.00 | 100.00 |

Table 7: Multilingual benchmark with additional similarity measures. Left block is with random sampling, and right block is FAISS sampling.

| | Random | | | FAISS | | |
|---|---|---|---|---|---|---|
| Layer | Dot product | Norm | ContraSim_norm | Dot Product | Norm | ContraSim_norm |
| 1 | $48.93 \pm 7.07$ | $68.67 \pm 10.86$ | $89.39 \pm 14.46$ | $15.00 \pm 6.41$ | $20.42 \pm 9.32$ | $15.76 \pm 6.56$ |
| 2 | $31.71 \pm 3.31$ | $74.19 \pm 9.73$ | $85.40 \pm 18.00$ | $20.14 \pm 2.75$ | $21.16 \pm 11.43$ | $18.83 \pm 11.19$ |
| 3 | $49.32 \pm 6.53$ | $83.92 \pm 13.12$ | $85.68 \pm 18.88$ | $13.00 \pm 4.50$ | $29.42 \pm 22.99$ | $26.36 \pm 16.01$ |
| 4 | $29.60 \pm 2.44$ | $99.61 \pm 0.41$ | $96.81 \pm 5.04$ | $16.20 \pm 4.01$ | $57.98 \pm 15.68$ | $28.79 \pm 9.80$ |
| 5 | $99.75 \pm 0.29$ | $99.86 \pm 0.33$ | $99.86 \pm 0.23$ | $82.17 \pm 7.36$ | $82.39 \pm 7.73$ | $74.04 \pm 7.11$ |
| 6 | $99.75 \pm 0.35$ | $99.84 \pm 0.27$ | $99.92 \pm 0.13$ | $83.38 \pm 7.70$ | $88.24 \pm 6.15$ | $77.00 \pm 6.68$ |
| 7 | $99.52 \pm 0.77$ | $99.85 \pm 0.29$ | $99.92 \pm 0.13$ | $89.72 \pm 5.57$ | $89.23 \pm 6.98$ | $78.21 \pm 6.77$ |
| 8 | $99.93 \pm 0.13$ | $99.89 \pm 0.15$ | $99.94 \pm 0.11$ | $93.49 \pm 4.07$ | $89.70 \pm 6.42$ | $81.97 \pm 6.63$ |
| 9 | $99.61 \pm 0.38$ | $99.76 \pm 0.40$ | $99.91 \pm 0.17$ | $82.48 \pm 9.32$ | $84.85 \pm 8.57$ | $81.37 \pm 6.53$ |
| 10 | $96.64 \pm 2.46$ | $99.38 \pm 0.58$ | $99.89 \pm 0.15$ | $55.02 \pm 15.42$ | $81.43 \pm 9.77$ | $80.59 \pm 6.83$ |
| 11 | $87.04 \pm 8.13$ | $98.40 \pm 1.25$ | $99.83 \pm 0.31$ | $29.62 \pm 13.82$ | $82.20 \pm 9.46$ | $80.74 \pm 7.08$ |
| 12 | $76.86 \pm 15.23$ | $87.25 \pm 12.39$ | $99.73 \pm 0.36$ | $25.75 \pm 26.55$ | $50.11 \pm 32.55$ | $80.24 \pm 6.63$ |

Similar to PWCCA, SVCCA requires that the number of examples is larger than the vector dimension, thus we could only evaluate it in the layer prediction benchmark. All other similarity measures were evaluated with all evaluations - the layer prediction benchmark, the multilingual benchmark and the image-caption benchmark.

Table 6 shows layer prediction benchmar results. We can observe that SVCCA achieves slightly better results than PWCCA, and lower than CKA and ContraSim. Both dot product and norm achieve low accuracies. ContraSim_norm achieves same or better results than ContraSim.

Table 8: Image–caption benchmark results for additional similarity measures, on 4 different model pairs.

| Vision Model | | ViT | | ConvNext | |
|---|---|---|---|---|---|
| Language Model | | BERT | GPT2 | BERT | GPT2 |
| Random | Dot Product | 6.32 | 6.32 | 11.39 | 8.86 |
|  | Norm | 6.32 | 7.59 | 15.19 | 7.59 |
|  | ContraSim_norm | 100 | 100 | 100 | 100 |
| FAISS | Dot Product | 5.06 | 2.53 | 7.59 | 6.32 |
|  | Norm | 5.06 | 5.06 | 6.32 | 10.12 |
|  | ContraSim_norm | 93.67 | 98.73 | 81.03 | 93.67 |

Multilingual benchmark results, summarized in Table 7 show that both dot product and norm achieve better results than CKA, although achieve low results under layer prediction and image-caption benchmarks. This emphasizes the importance of multiple evaluations for similarity measures. Compared to ContraSim, both methods achieve lower results. ContraSim_norm achieves lower results compared to ContraSim, under both random and FAISS sampling.

Image-caption benchmark results, summarized in Table 8, show that under both random sampling and FAISS sampling dot product and norm achieve low accuracy. Under random sampling, ContraSim_norm achieves perfect accuracy, while using FAISS sampling shows slight degradation compared to ContraSim.

