# OpenReview forum: "ContraSim -- A Similarity Measure Based on Contrastive Learning"
_ICLR.cc/2023/Conference — Submitted to ICLR 2023_

### Official Review · Reviewer_1R2q · 2022-10-22

**Confidence:** 4
**Correctness:** 4
**Technical Novelty And Significance:** 2
**Empirical Novelty And Significance:** 4
**Recommendation:** 8

**Clarity, Quality, Novelty And Reproducibility:**

The paper is excellent in clarity, and reproducibility.

The novelty is above par, but a simple extension of existing ideas.
The quality can be improved with deeper analysis, as mentioned in weaknesses.

**Strength And Weaknesses:**

Strengths:
- Paper is well-written, easy to follow.
- Introduction of two benchmarks adds a different perspective to this research area compared to literature
- Proposed method outperforms baselines by a significant margin

Weaknesses:
- The analyses in the paper can dive deeper into the problem space. How much does ContraSim generalize? Can you use the same  ContraSim model across all benchmarks instead of training a separate one for each?
- It would be good to discuss the trade-offs introduced by ContraSim. E.g. we need to access to labeled training data. We need different models for each domain.
- The paper does not discuss the implications of this research. Why is learning based measure useful? What did we learn that's new?

**Summary Of The Paper:**

The paper proposes a learned measure of representation similarity, ContraSim, as opposed to analytical ones like centered kernel alignment (CKA) and canonical correlation analysis (CCA). ContraSim is specific to a task (training distribution), and uses a simple MLP to project the representation to a different subspace. Experiments show that contrastive loss is a key component to achieving good results.

Paper also introduces two additional benchmarks based on multi-lingual similarity and image-caption similarity.  ContraSim performs well on these tasks compared to baselines.

**Summary Of The Review:**

I think the paper is above the acceptance bar for ICLR. I would like to see some additional analyses and discussion in the final version of the paper.

---

> ### Author Response · Authors · 2022-11-16
> **Response to Reviewer 1R2q**
>
> We thank the reviewer for their comments and feedback. We appreciate that the reviewer found the results to be good and valuable and that the paper was clearly written. Below we discuss the point the reviewer raised:
>
> > Can you use the same ContraSim model across all benchmarks instead of training a separate one for each?
>
> We have done transfer learning experiments, for example across datasets (in the layer prediction benchmark) and across models (in the image caption benchmark). However, We wanted ContraSim to be a task-specific method and not a general-purpose one. A general-purpose similarity measure will probably perform less well on each task and will require a large training time for a single encoder as it will need to be trained on several datasets. We believe that a task-specific similarity measure is a better option. We added a discussion of this issue in the Related Work section.
>
> > It would be good to discuss the trade-offs introduced by ContraSim.
>
> Thank you for drawing our attention. We have expanded our discussion of the trade-offs introduced by ContraSim. In particular, we added to the final draft a discussion regarding the need for a training dataset compared to the existing method. However, regarding the fact that ContraSim uses a different model for each task, we consider this as an advantage and not a drawback, for the reasons discussed above. Other drawbacks we discussed are regarding comparing representations with a different dimension, with a proposed solution, and the method training time compared to closed form similarity measures. We added this discussion in the Conclusion section.
>
> > The paper does not discuss the implications of this research.
>
> Since ContraSim is a learnable similarity measure with better performance on all evaluations compared to existing methods, it will be a better fit for interpretability of neural networks, providing more reliable results. We added a discussion about new insights we get from ContraSim in multilingual and image-caption benchmarks results in section 6 (Experiments).
>
> From the multilingual benchmark, Table 7 in the paper, with FAISS sampling we see that there is a much greater difference in accuracy between shallow and deep layers in ContraSim compared to previous similarity measures. This difference indicates a difference in similarity between representations in shallow and deep layers that wouldn’t be noticed with previous similarity measures. In addition, in the image-caption benchmark, Figure 3 in the paper, using CKA, pairs with the BERT model achieve much higher accuracy than pairs with GPT2 models. From that we might infer that BERT model representations are closer to vision model representations. However, ContraSim contradicts that as all pairs achieve high accuracy. From that we learn that both BERT and GPT2 representations are close to computer vision models representations.
> Moreover, being a learnable similarity measure, we believe ContraSim may help train models with similarity objectives. We discussed this in the conclusion section.

---

### Official Review · Reviewer_m4jx · 2022-10-24

**Confidence:** 3
**Correctness:** 3
**Technical Novelty And Significance:** 2
**Empirical Novelty And Significance:** 3
**Recommendation:** 6

**Clarity, Quality, Novelty And Reproducibility:**

Clarity: Good.

Quality: Good.

Novelty: Good.

Reproducibility: Fully reproducable.

**Strength And Weaknesses:**

Strength:
  - The paper is well-written and easy to follow.
  - The method is simple yet effective.
  - The empirical analysis is comprehensive.
  - The provided benchmark can benefit future research.
  - This paper provides a novel perspective in providing similarity measures compared to the existing closed-form ones.

Weakness:
  - As a learnable similarity measure, due to the randomness of model training, this metric cannot guarantee consistent results, which may limit its use cases and reliability to some extent.

**Summary Of The Paper:**

This paper introduces a parameterized similarity measure that shows significant improvement over existing closed-form similarity measures.

**Summary Of The Review:**

This paper provides a novel similarity measure based on contrastive learning, which greatly outperform existing closed-form measures in different scenarios.

---

> ### Author Response · Authors · 2022-11-16
> **Response to Reviewer m4jx**
>
> We thank the reviewer for their comments and feedback. We appreciate that the reviewer found the results to be novel, comprehensive, and valuable and that the paper was clearly written. Below we discuss the weakness the reviewer raised:
>
> >As a learnable similarity measure, due to the randomness of model training, this metric cannot guarantee consistent results, which may limit its use cases and reliability to some extent.
>
> We have taken this point to heart, and in all evaluations we added some randomness in-order to ensure the consistency of the results. In the layer prediction benchmark and the image caption benchmark, we trained and evaluated ContraSim on 5 different seeds and reported the mean and standard deviation of the results. As reported in the paper, the standard deviation was very low, ensuring the consistency of our method against randomness.
> In the multilingual benchmark, we trained the encoder on a specific language pair, from all 20 possible pairs we used, and evaluated it against all the 19 remaining language pair options. We then reported mean accuracy and standard deviation over all choices of pairs. From the reported results we can observe that in most evaluations, ContraSim’s standard deviation was lower than all other similarity measures we evaluated.
>
> Please let us know if you have any other questions or concerns. If you are satisfied with our response to the weakness above, we would appreciate it if you consider revising your score accordingly.

---

> > ### Author Response · Authors · 2022-12-01
> > **Response to Reviewer m4jx**
> >
> > We would like to kindly thank again the reviewer for their comments and feedback. In light of the upcoming discussion deadline, we would like to know whether the reviewer saw our response and whether it addresses their concerns.

---

### Official Review · Reviewer_rucx · 2022-10-24

**Confidence:** 4
**Correctness:** 2
**Technical Novelty And Significance:** 1
**Empirical Novelty And Significance:** 2
**Recommendation:** 3

**Clarity, Quality, Novelty And Reproducibility:**

The clarity and the writing could be improved. Some spelling errors should be corrected.
The technical novelty is limited as the proposed model makes use of existing methods of contrastive learning.
Reproducibility is constrained by the use of retrieved examples, but when train data are provided by the authors, this problem is mitigated.

**Strength And Weaknesses:**

Strengths:
- Evaluation of the created metric on different types of data.

Weaknesses:
- Clarity of the work and its goals.
- Limited technical contribution.
- Evaluation only on three benchmarks, two of which are created.
- Comparison with existing similarity metrics is limited.


**Summary Of The Paper:**

The paper proposes a similarity metric learned from similar and dissimilar examples using a loss inspired by contrastive learning. The metric is tested on three benchmarks (i.e., layer prediction, multilingual language modeling, image captioning), among which are two new benchmarks proposed by the authors on which they test the metric.


**Summary Of The Review:**


When defining a new similarity metric, one should first define its theoretical properties and evaluate these properties empirically with data. These important parts are lacking in the paper.

The motivation of the work is not clear. Is it the purpose to learn an encoder based on a contrastive loss or a similarity metric that can be used in and transferred to all kinds of applications?

What would be the effect if another similarity metric than cosine similarity would be used to compare representations during training?

The novelty of the work seems limited. Learning representations from positive and negative examples based on a contrastive loss and using these to create a semantic space where similarity is measured has been done for almost a decade. The number of baselines and comparisons with other similarity metrics seem limited.

The authors rely on the FAISS library for similarity search of train examples. What is the influence of this initial search and the underlying methods that FAISS uses on the quality of the examples used in the training and consequently on the quality of the learned metric?

---

> ### Author Response · Authors · 2022-11-16
> **Response to Reviewer rucx**
>
> We thank the reviewer for their comments and feedback. Below we discuss the points the reviewer raised:
>
> > What would be the effect if another similarity metric than cosine similarity would be used to compare representations during training?
>
> Thank you for this proposal. While our results confirm that cosine similarity leads to an effective similarity measure in ContraSim, we performed several experiments to answer your questions. We defined a diss-similarity measure as the $L_2$ norm of the distance between two normalized representations. For a batch of representations, the batch diss-similarity is the mean of the pairwise norm, i.e.: $Dis\_{\text{norm}}(f_1, f_2) = \text{mean}(\|\|(f_1 / \|\|f_1\|\| -  f_2/\|\|f_2\|\|)$.
> Since this is a dis-similarity measure, we defined the similarity measure as $ S(f_1, f_2) = 1 - Dis\_{\text{norm}}(f_1, f_2)$.
>
> We then ran the following evaluations:
> 1. Layer prediction benchmark with two datasets: Penn TreeBank and WikiText. In all evaluations, the results are the same or better than the results of the original ContraSim.
> 2. Multilingual benchmark with random sampling and FAISS sampling. Under random sampling results are very close to the original ContraSim. Using FAISS sampling shows a degradation in accuracy compared to ContraSim.
> 3. Image-caption benchmark with 4 different model pairs for computer vision and language models. Under this benchmark we see 100% accuracy under random sampling across all pairs; however, using FAISS sampling shows slight degradation in accuracy compared to the original ContraSim.
>
> We include these results in the final draft. They can be found in Appendix A.5 in Tables 6, 7 and 8.
>
> > Evaluation only on three benchmarks, two of which are created.
>
> In the field of designing similarity measures, the layer prediction benchmark is a common benchmark [1, 2], sometimes the only one, like in the CKA paper [2].
> In order to obtain a better evaluation of our method, we proposed two other evaluations: the multilingual benchmark and the image-caption benchmark. We believe those benchmarks constitute an in-depth examination of our proposed method, and we hope they will also be beneficial for future research on similarity measures.
>
> > Comparison with existing similarity metrics is limited.
>
> We compared our method with the current most popular similarity measures: CKA [2] and PWCCA [3]. Following your comment, we added evaluations of three more similarity measures: SVCCA [4], dot product and L2 norm between the difference of the normalized representations (norm). Similar to PWCCA, SVCCA requires that the number of examples is larger than the vector dimension, thus we could only evaluate it in the layer prediction benchmark.
>
> We ran the following evaluations:
> 1. Layer prediction benchmark with two datasets: Penn TreeBank and WikiText. In all evaluations, the accuracy results of all new methods are much lower than those of ContraSim.
> 2. Multilingual benchmark with random sampling and FAISS sampling: Under both random and FAISS sampling. ContraSim shows superior results over dot product and norm. Using a FAISS-based sampler further extends the gaps. It's interesting to see that both dot product and norm achieve better results than CKA under this benchmark. CKA is considered a better similarity measure in the literature and dot product and norm dramatically fail under layer prediction and image-caption benchmarks. This emphasizes the importance of multiple evaluations for similarity measures.
> 3. Image-caption benchmark with 4 different model pairs for computer vision and language models: Both dot product and norm poorly perform in this benchmark, both with random or FAISS sampling.
>
> The results can be found under Appendix A.5 in Tables 6, 7 and 8.
>
> >The authors rely on the FAISS library for similarity search of train examples. What is the influence of this initial search and the underlying methods that FAISS uses on the quality of the examples used in the training and consequently on the quality of the learned metric?.
>
> FAISS library is only used for evaluation in the multilingual benchmark and the image-caption benchmark. FAISS was not part of  ContraSim training procedure in any evaluation, thus it does not have any effect on ContraSim learned metric. However, ContraSim still achieves high accuracy while facing challenging examples using FAISS in the evaluation step.
>
> > The motivation of the work is not clear.
>
> Thank you for drawing our attention. In the introduction's first paragraph we discussed why similarity measures are important for the interpretability of neural networks. In order to emphasize our motivation, we added to the second paragraph motivation for why we need a new similarity measure, and what motivated us to propose ContraSim.
>
> If this helps address your feeling that the paper is a “reject”, we would be grateful if you considered revising your score.

---

> > ### Author Response · Authors · 2022-11-16
> > **References**
> >
> > References:
> >
> > [1] Ding, Frances, Jean-Stanislas Denain, and Jacob Steinhardt. "Grounding Representation Similarity Through Statistical Testing." Advances in Neural Information Processing Systems 34 (2021): 1556-1568.
> >
> > [2] Kornblith, Simon, et al. "Similarity of neural network representations revisited." International Conference on Machine Learning. PMLR, 2019.
> >
> > [3] Morcos, Ari, Maithra Raghu, and Samy Bengio. "Insights on representational similarity in neural networks with canonical correlation." Advances in Neural Information Processing Systems 31 (2018).
> >
> > [4] Raghu, Maithra, et al. "Svcca: Singular vector canonical correlation analysis for deep learning dynamics and interpretability." Advances in neural information processing systems 30 (2017).

---

> > > ### Author Response · Authors · 2022-12-01
> > > **Response to Reviewer rucx**
> > >
> > > We would like to kindly thank again the reviewer for their comments and feedback. In light of the upcoming discussion deadline, we would like to know whether the reviewer saw our response and whether it addresses their concerns.

---

> > ### Comment · Reviewer_rucx · 2022-12-03
> > **Comment of reviewer on novelty**
> >
> > I thank the authors for giving additional clarifications and adapting the paper. However, I am not yet convinced that the paper has sufficient novelty. Can the authors clarify the difference of their idea with related ideas such as used in biased discriminant analysis (e.g., discussed in https://ieeexplore.ieee.org/stamp/stamp.jsp?tp=&arnumber=4469875), or the use of a similarity metric by learning semantic feature spaces based on a contrastive loss (e.g., Karpathy et al. NIPS 2014), the latter model being used in many variations and with refinements of the contrastive loss?

---

> > > ### Author Response · Authors · 2022-12-05
> > > **Response to Reviewer rucx**
> > >
> > > We thank the reviewer for thoroughly reading our paper and referring us to additional relevant papers. In both papers referred by the reviewer, their proposed methods use contrastive learning-like methods for the final purpose of data retrieval. They used contrastive learning to create a ranking of samples given a query. However, we developed ContraSim for a different goal.
> > > What interested and motivated us to develop ContraSim was to find a similarity measure as good as possible from considerations of analysis and interpretability of deep neural networks. To determine how good a similarity measure is we use benchmarks that have similarities to information retrieval. However, this is not the main goal of our method. We aim for a similarity measure that will assist in the analysis of deep neural networks.
> > > Indeed, in our updated paper, we added insights regarding interpretability results we found using ContraSim that were not found using prior similarity measures.

---

### Official Review · Reviewer_xBqQ · 2022-10-24

**Confidence:** 4
**Correctness:** 3
**Technical Novelty And Significance:** 2
**Empirical Novelty And Significance:** 2
**Recommendation:** 5

**Clarity, Quality, Novelty And Reproducibility:**

The paper is clear and easy to read. And it should be easily reproducible once the authors' code is made public, as they have promised to make it upon publication.

As described above, the lack of novelty is the major shortcoming of this work. And the missing empirical evaluations against highly similar contrastive learning methods critically impact the quality of the work as well.


**Strength And Weaknesses:**


The strength of this paper is that it emphasizes the importance of negative examples in training similarity functions and shows that they are critical in contrastic learning. I also liked that the authors experimented on embeddings produced by SOTA models like RoBerta and ViT.

However, the weakness of the paper is that using negative pairs is very common in contrastive learning, despite the paper's claiming the opposite. For instance, in SimCLR (which this paper cites), the objective maximizes the dot products of should-be positive pairs and minimizes the dot products of negative pairs (see equation (1) in the SimCLR paper).

Likewise, well known contrastive learning criteria like noise contrastive estimation (NCE), InfoNCE and triplet loss seek to minimize dot products of negative pairs relative to positive pairs.

These contrastive learning objectives need to be compared against because they are all highly similar to the proposed Contrasim.

In particular, equation (3) Contrasim can be seen as maximizing the log-odds probability of positive versus negative pairs if we assume there is a normalizing factor "Z" which has been divided out from both the numerator and denominator:

  eq (3) = \sum_i log [ \sum_p\~i [ exp(ai . ap)/Z ] / [ \sum_n!\~i exp(ai . an)/Z ].

where "\~" denotes the similarity relation and "!\~" denotes the not-similar relation.

SimCLR can be seen as maximizing the very similar odds ratio:

  \sum_i \sum_p\~i log { [ exp(ai . ap)/Z ] / [ \sum_n!\~i exp(ai . an)/Z ] },

so it is not clear if Contrasim presents a notable advance, especially in absence of empirical comparisons and/or theoretical justifications.

**Summary Of The Paper:**


This paper proposes a new contrastive learning objective for learning representations called Contrasim. Contrasim aims to learn representations that maximize inner-products between "should-be-similar" inputs and minimize them between "should-be-dissimilar" inputs. Contrasim's claim to novelty is that it adds the additional minimization objective for the "should-be-dissimilar" representation pairs whereas other representation learning objectives like project-weighted CCA (PWCCA) and centered kernel alignment (CKA) do not.

The authors test Contrasim on 3 representation learning tasks:
1) Determining which encodings of sentences from two different languages are translations of each other.
2) Determining which sentence and image encodings are (caption,image) pairs.
3) For neural networks with the same architecture and training data, but different initialization, determine which layers of the neural networks correspond.

Experimental results show that Contrasim dramatically outperforms the chosen baselines: CKA, PWCCA, deep dot-product and deep CKA; and the performance gap increases when using hard negative mining with the FAISS library.

Last, low dimensional projection plots show that Contrasim improves the proximity of should-be-similar inputs relative to the original representations and relative to the aforementioned baselines.

**Summary Of The Review:**

While Contrasim is a moderately intriguing variant of existing contrastive learning losses, it is too similar to existing methods like SimCLR and NCE and not well enough distinguished from them to warrant acceptance at ICLR.

---

> ### Author Response · Authors · 2022-11-16
> **Response to Reviewer xBqQ**
>
> We thank the reviewer for their comments and feedback. We appreciate that the reviewer found the results to be significantly better than other methods, and that the paper was clearly written. However, we believe the reviewer misunderstands the purpose of the work. Our work does not suggest a new representation learning method, but rather a new learnable similarity measure, which we name ContraSim.
>
> Below we discuss some of the points the reviewer raised:
>
> > This paper proposes a new contrastive learning objective for learning representations called Contrasim.
>
> ContraSim is not a contrastive learning objective, but rather a new measure of similarity based on contrastive learning. Our work contributes to the field of analyzing neural network representations via similarity measures. As part of our method, it learns projections of representations. However, those projections are only used for the purpose of computing similarity and not for representation learning downstream tasks.
>
> > The authors test Contrasim on 3 representation learning tasks.
>
> We evaluated ContraSim on 3 similarity measures evaluation benchmarks, and not representation learning tasks. Those evaluations are a sanity check of similarity measures in order to evaluate their effectiveness against a hypothesis we have regarding the similarity of different objects (each evaluation checks a different hypothesis).
>
> > These contrastive learning objectives need to be compared because they are all highly similar to the proposed Contrasim.
>
>
> Our work uses ideas from contrastive learning but is not a competitor of it. ContraSim and contrastive learning share similar ideas but are from different domains. Contrastive learning is used for representation learning, whereas ContraSim is used to measure similarity between representations. Contrastive learning is not used for similarity measure, and neither InfoNCE nor triplet loss, thus the proposed comparison is not valid.
>
> We hope those clarifications will be beneficial for the paper's understanding. If this helps address your feeling that the paper is a “reject”, we would be grateful if you considered revising your score.

---

> > ### Comment · Reviewer_xBqQ · 2022-11-29
> > **Reply to authors reply**
> >
> > To the authors, many thanks for clarifying your contributions. I now understand them better. I will revise my original review scores accordingly.
> >
> > Reformulating my original summary, the authors compare their method for learning a *cross-domain* similarity function to traditional kernal alignment methods like (PW)CCA and CKA, and also to two deep learning variants: DeepCKA and DeepDot.
> >
> > ContraSim uses the same objective in training as previous methods (SimCLR and InfoNCE) but applies it to learning a similarity among pre-existing embeddings rather than learning a similarity on the original inputs.
> >
> > > "ContraSim is not a contrastive learning objective, but rather a new measure of similarity based on contrastive learning.
> >  ...
> > Our method uses a trainable encoder, which first maps representations to a
> > new space and then measures the similarity of the projected representations.
> > ...
> > In order to obtain a similarity score between 0 and 1, we first apply L2 normalization to the encoder outputs: z1 = eθ(a1)/∥eθ(a1)∥ (and similarly for a2). Then their similarity is calculated as: s(z1, z2) where s is a simple closed-form similarity measure for two vectors. Throughout this work we use dot product for s."
> >
> > However, I stand by my original sentiment regarding lack of originality,as output vectors produced by one model can always be used as inputs to another. So I do not see a material difference between using ContraSim and applying SimCLR (or InfoNCE) to a different data source, which happens to consist of pre-trained embeddings. This is to say, one person's metadata is another person's data.
> >
> > > Contrastive learning is not used for similarity measure, and neither InfoNCE nor triplet loss, thus the proposed comparison is not valid."
> >
> > Here, I respectfully disagree. The dot-product (or cosine) scores induced by InfoNCE and triplet loss are frequently used as similarity measures in information retrieval and collaborative filtering applications. This is why vector search databases such as FAISS, used in your experiments, are useful.  Word2Vec -- also contrastive learning -- is useful because the dot products among representations are useful as a measure of word similarity.
> >
> > You are correct in that these methods were not used to learn cross domain similarities in the original articles. But they have since been used to do so (Zeng et al. NeurIPS 2021; Lin et al. COLING 2022).

---

> > > ### Author Response · Authors · 2022-11-30
> > > **Response to Reviewer xBqQ**
> > >
> > > We thank the reviewer for thoroughly reading our paper and referring us to additional relevant papers. What interested and motivated us to develop ContraSim was to find a similarity measure as good as possible from considerations of analysis and interpretability of deep neural networks. To determine how good a similarity measure is we use benchmarks that have similarities to information retrieval. However, this is not the main goal of our method. We aim for a similarity measure that will assist in the analysis of deep neural networks.
> > >
> > > We see that you wrote that you will revise your original score, but there is no change in the score. We hope this clarification will contribute to the paper's understanding and we would be grateful if you considered revising your score.

---

> > > > ### Comment · Reviewer_xBqQ · 2022-11-30
> > > > **Revised scores**
> > > >
> > > > Paper3112 authors, I just revised my scores upward

---

> > > > > ### Author Response · Authors · 2022-12-01
> > > > > **Appreciation for the reviewr**
> > > > >
> > > > > We kindly thank the reviewer for updating our score and investing time to review our paper.

---

### Author Response · Authors · 2022-12-13
**General Response and Summary of Updates to Manuscript**

We would like to thank the reviewers for their comments and feedback. First, we provide a high-level summary of the changes that we've made to the draft to address your feedback:
1. Added a new evaluation of ContraSim with a different similarity metric - the $L_2$ norm of the distance between two normalized representations.

2. Added evaluation of additional similarity measures - SVCCA, dot-product, and L2 norm between the difference of the normalized representations.

3. Added paragraphs regarding the use of ContraSim for the interpretability of neural networks. We added a discussion about new insights we get from ContraSim in multilingual and image-caption benchmark results.

4. Expanded our discussion of the trade-offs introduced by ContraSim.

Second, we summarize our key contributions:
1. We introduce a new similarity measure – ContraSim. Inspired by contrastive learning, it uses positive and negative sets to train an encoder that maps representations to the space where similarity is measured. In our evaluations, ContraSim reveals new interesting insights that were not found using previous similarity measures.

2. We propose two new benchmarks for the evaluation of similarity measures: the multilingual benchmark and the image–caption benchmark.

3. We show that ContraSim outperforms existing similarity measures in all benchmarks, and maintains high accuracy even when faced with more challenging examples.

We would like to point out reviewers m4jx and 1R2q for mentioning the novelty of our proposed method. Regarding reviewer rucx who was concerned by the novelty of our method, we answered his/her review and tried to address his/her concerns, however, no additional feedback was provided.

---

### Decision · Program_Chairs · 2023-01-20

**Decision:**

Reject

**Justification For Why Not Higher Score:**

lack of strong support in the reviews, the limited overall scope of learning a similarity function

**Justification For Why Not Lower Score:**

N/A

**Metareview: Summary, Strengths And Weaknesses:**

The paper proposes an empirically successful similarity measure called ContraSim. It takes fairly large positive and negative sets as input, trains a separate encoder using contrastive loss, and then evaluate the final similarity as a dot product of these encoded representations. This paper only learns the similarity function without modifying the underlying representations themselves, but instead the parameters are in the trainable encoder.

There was some confusion among the reviewers that this work is learning representations, which it is not. The authors addressed this adequately and some reviewers agreed. I suspect a source of this confusion is that in eq 3), a_i was used instead of z_i, which makes it seem like the original representation a_i was being learned. The authors were not very clear on the benefits of learning this separate encoder z=e(a), though since e is just a 2 layer NN and it seems easy to imagine some benefits over relearning the whole representation for each layer.  It is also perhaps not surprising that additional negative examples can be helpful for these tasks, but still showing this is a clear contribution.

The paper is clear in that it is a trainable similarity measure, and it seems to do well on the advertised goal by outperforming baselines by a clear margin. 2 reviewers liked the paper and many good comments (confusing points about FAISS and SimCLR, further impact) were raised by the negative reviewers.  The main quantitative results are on artificial layer prediction task or pick-a-neighbor binary contrast task, it is unclear what are the impacts of these clear improvements. The paper can be improved by addressing these confusion points and perhaps expanded to include results on downstream tasks.